# Molecular Analysis of Pancreatic Cyst Fluid for the Management of Intraductal Papillary Mucinous Neoplasms

**DOI:** 10.3390/diagnostics12112573

**Published:** 2022-10-24

**Authors:** Ronald C. Turner, Jared T. Melnychuk, Wei Chen, Daniel Jones, Somashekar G. Krishna

**Affiliations:** 1College of Medicine, The Ohio State University, Columbus, OH 43210, USA; 2Department of Pathology, The Ohio State University, Columbus, OH 43210, USA; 3Comprehensive Cancer Center, The Ohio State University, Columbus, OH 43210, USA; 4Department of Gastroenterology, The Ohio State University, Columbus, OH 43210, USA; 5Division of Gastroenterology, Department of Internal Medicine, Ohio State University Wexner Medical Center, Columbus, OH 43210, USA

**Keywords:** cyst fluid, IPMN, molecular analysis, pancreatic cancer, pancreas

## Abstract

Pancreatic cancer is one of the most lethal human cancers. Early detection and diagnosis of precursor lesions for pancreatic malignancy is essential to improve the morbidity and mortality associated with this diagnosis. Of the cystic precursor lesions, branch duct intraductal papillary mucinous neoplasm (IPMN) is the most frequently identified lesion and has a wide range of malignant potential. Currently, Carcinogenic embryonic antigen (CEA) levels in the cyst fluid and cytology are the two most often utilized tools to diagnose these lesions; however, their diagnostic and risk stratification capabilities are somewhat limited. Within the last decade, the use of endoscopic ultrasound-guided fine-needle aspiration has opened the door for molecular analysis of cystic fluid as an option to enhance both the diagnosis and risk stratification of these lesions. The first step is to differentiate branch duct IPMNs from other lesions. *KRAS* and *GNAS* alterations have been shown to be accurate markers for this purpose. Following cyst type identification, mutational analysis, telomere fusion, microRNAs, long non-coding RNA, and DNA methylation have been identified as potential targets for stratifying malignant potential using the cystic fluid. In this review, we will examine the various targets of cyst fluid molecular analysis and their utility in the diagnosis and risk stratification of branch duct IPMNs.

## 1. Introduction

With the advancement of medical imaging techniques, pancreatic cystic lesions (PCLs) are being incidentally discovered at increasing rates. Branch duct intraductal papillary mucinous neoplasms and mucinous cystic neoplasms (MCNs) are the most frequently identified precancerous cystic lesions of the pancreas. Branch duct IPMN is a grossly visible (typically >5 mm) intraductal epithelial neoplasm of mucin-producing cells, arising in the smaller, secondary pancreatic ducts. Branch duct IPMNs have a variable rate of malignant transformation [1]. It is important to accurately risk-stratify branch duct IPMNs into those with and without advanced neoplasia (high-grade dysplasia or invasive carcinoma) [2]. While there is surgical overtreatment of branch duct IPMNs at one end, where nearly 50% of those resected are found to have low-grade dysplasia, there is also a risk of missing branch duct IPMNs with advanced neoplasia at the other [3,4]. Pancreatic surgery has an associated 30% risk of morbidity and a 1–2% risk of postoperative mortality [5].

Currently, pancreatic cyst fluid analysis with carcinogenic embryonic antigen (CEA) and cytology is recommended to differentiate between mucinous and non-mucinous cysts but is not helpful for risk stratification of branch duct IPMNs [6]. Molecular analysis of pancreatic cyst fluid obtained via fine needle aspiration (FNA) has emerged in recent years as a viable option to determine the malignant potential of branch duct IPMNs [7]. One of the major advantages of pancreatic cyst fluid molecular analysis is the relatively small fluid volume needed for testing when compared to CEA and cytology. Inadequate cytology sampling frequently occurs due to a lack of sufficient volume or low cellularity of the fluid component within the PCL, whereas DNA analysis only requires a 0.4 mL sample [8]. An alternative method for identifying and potentially risk stratifying IPMNs and other pancreatic lesions is through the needle biopsy (TTNB). TTNB utilizes endoscopic ultrasound (EUS) and passes micro biopsy forceps through the needle to obtain a sample of the cyst wall [9], which can then be analyzed for diagnosis. While the utility of TTNB for mutational analysis has been well-documented [10], less research has been pursued regarding the feasibility of obtaining TTNB samples and the diagnostic yield, sensitivity, and specificity of the method. Through the use of TTNB, Rift et al., reported a sensitivity and specificity for the diagnosis of IPMN in the same range as cyst fluid analysis [11]. The benefit of using TTNB is that it has fine diagnostic capability. However, there is a large risk (as high as 10%) of pancreatitis and bleeding associated with this procedure, limiting its use [11]. Therefore, TTNB is a good option for patients where the necessity of an accurate diagnosis outweighs the risks. Molecular analysis of cystic fluid provides an option that sustains the high level of diagnostic accuracy demonstrated by TTNB with a lower risk of adverse events. 

Cyst fluid can be useful both for the initial diagnosis of branch duct IPMN, as well as for risk stratification of the lesions after diagnosis. Traditionally, surgical resection is pursued based on clinical suspicion, radiologic data, and CEA level. CEA has been shown to be more specific for mucinous lesions, however, NGS is much more sensitive (86% vs. 57%) [12]. This brings to light the question of how much molecular analysis contributes to medical decision-making in the management of pancreatic cystic lesions. In a study by Arner et. al, the addition of molecular analysis changed management in about a quarter of patients, however it changed management in 40% of cases when CEA levels were indeterminant (*p* < 0.05) [13]. This review will focus on cyst fluid analysis in branch duct IPMN and its relevance in the management of pancreatic cysts. Some of the studies referenced examine other mucinous neoplasms in addition to branch duct IPMNs, such as main-duct IPMNs and mucinous cystic neoplasms, however, molecular alterations are similar across mucinous lesions. Molecular studies have shown that branch duct IPMNs frequently harbor mutations in *KRAS* and/or *GNAS* [14]. In addition, *BRAF* mutations can be seen with *GNAS* when *KRAS* is not detected, suggesting that dysregulated *RAS-MAPK* signaling is common to neoplastic initiation in all branch duct IPMN cases [15]. However, the range of genetic alterations occurring subsequently likely contribute to differential clinical behavior. Within the last decade, studies on pancreatic cyst fluid to identify those molecular progression events that could be used for risk-stratification have included: gene mutation analysis, copy number analysis to detect loss of heterozygosity (LOH) in tumor suppressor genes, telomere dysfunction, gene fusions, microRNA (miRNA), and DNA methylation [15,16,17,18,19,20]. Further identification of biomarkers capable of diagnosing and predicting the malignant transformation of branch duct IPMNs is necessary to help reduce the number of patients undergoing unnecessary preemptive pancreatic resections.

## 2. The Diagnostic Role of *KRAS* and *GNAS* Mutations in Branch Duct IPMN Classification

In recent years, there have been some important studies utilizing next-generation sequencing (NGS) analysis in the diagnosis of branch duct IPMNs. *KRAS* and *GNAS* are the most commonly mutated oncogenes contributing to pancreatic cancer, and these mutations tend to occur early on in branch duct IPMN tumorigenesis (Table 1) [21]. It is well known that mucinous lesions are far more likely to harbor malignant potential, therefore accurate distinction between mucinous and non-mucinous cysts is a crucial first step in the diagnosis. A small 2021 study identified KRAS/GNAS testing to have a 94.7% sensitivity and 100% specificity for mucinous differentiation, thus implicating its use as a single diagnostic test [22]. GNAS alone has been observed exclusively in BD-IPMNs, thus making fluid GNAS a useful marker in further differentiation following mucinous classification [23]. In cysts with a non-mucinous CEA level, analysis of KRAS or GNAS alone has been shown to reclassify 71% of cysts as BD-IPMN, thus improving our ability to detect cysts that need further investigation [12]. However, analysis of KRAS and GNAS in combination has been shown to be significantly more accurate in the diagnosis of IPMN [24]. A large meta-analysis of 785 total lesions suggested that KRAS and GNAS together is extremely useful in the diagnosis of BD-IPMN [24]. KRAS and GNAS analysis achieved a sensitivity, specificity, and diagnostic accuracy of 94%, 91%, and 97% [24]. In a separate study utilizing combination analysis, the specificity and sensitivity to diagnose branch duct IPMNs reached 98% and 84%, respectively [25]. A large 2016 study even showed *KRAS* and/or *GNAS* analysis to have 100% specificity and sensitivity for branch duct IPMN identification [26]. 

Molecular analysis can be of immense diagnostic value, especially if it is used in conjunction with other tests. In 2017, Kadayifci et al., assessed the diagnostic value of adding *GNAS* to *KRAS* and CEA testing of pancreatic cyst fluid. The combination of all three markers led to a significantly improved diagnostic accuracy for branch duct IPMN when compared to single tests (*p* < 0.05) [27]. However, *KRAS* and/or *GNAS* detection alone in pancreatic cyst fluid has been found to be highly sensitive and specific (100% and 96%, respectively) for the diagnosis of branch duct IPMN [16]. This was uncovered in the premier study by Singhi et al., in 2018. In this study, molecular analysis significantly outperformed fluid viscosity and elevated CEA levels as diagnostic modalities. Additional mutations have been identified that contribute to branch duct IPMN carcinogenesis. Ren et al., showed that in mucinous PCLs without *KRAS* mutations, *BRAF* mutations with concurrent *GNAS* mutations supported an alternate mechanism of activation in the Ras pathway [15]. With continued research over the years, the sensitivity and specificity of the mutations discussed above have improved to a point that should give physicians confidence in their use as a clinical tool. 

## 3. Molecular Analysis for the Risk Stratification of Branch Duct IPMNs

Branch duct IPMNs are typically characterized as either low-grade dysplasia or advanced neoplasia (high-grade dysplasia or invasive carcinoma). It is vital to accurately risk stratify branch duct IPMNs to avoid unnecessary surgeries, but current guidelines are suboptimal. A multi-institutional study done in 2017 showed the diagnostic accuracy of Fukuoka, AGA, and ACR criteria to only be 49.8%, 75.8%, and 59.8%, respectively, for the detection of advanced neoplasia [28]. The various potential options for improving risk stratification of BD-IPMNs using molecular analysis of cyst fluid are discussed below (Table 2).

### 3.1. Multi-Gene Mutational Analysis of Cyst Fluid 

Over the last decade, there have been many studies performed with the goal of risk-stratifying PCLs. The 2009 PANDA study was fundamental in establishing this concept. They discovered that high amplitude *KRAS* mutations in conjunction with allelic loss had a specificity and sensitivity of 96% and 37%, respectively, for malignancy [29]. In 2015, Springer et al., uncovered that the best predictor of branch duct IPMN with advanced neoplasia was the existence of mutations in *SMAD4*, LOH in the chromosomal region of *RNF43* or *TP53*, or aneuploidy in any of the chromosomes 5p, 8p, 13q, 18q [30]. Notably, the use of molecular analysis in this study could have, in retrospect, saved 90% of patients with low-grade pathology from surgery. Additionally, the presence of *TP53*, *PIK3CA*, and/or *PTEN* has been shown to have a sensitivity and specificity of 91% and 97%, respectively, for branch duct IPMN with advanced neoplasia, and improved even further when combined with *KRAS* and/or *GNAS* alterations [26]. Studies done in the last five years have continued to identify crucial mutations that can be used for the risk stratification of branch duct IPMNs.

Recent studies have advocated for the use of NGS in conjunction with cytology and/or CEA to increase diagnostic accuracy and sensitivity for neoplastic cyst detection [15,31]. In 2017, Rosenbaum et al., reported mutations in *TP53, SMAD4*, *CDKN2A*, and *NOTCH1* to be exclusively present in carcinoma, but were of limited sensitivity when compared to cytology [32]. In a follow-up from their initial study, Singhi et al., adjusted their initial selection criteria to only include *GNAS* mutation allele frequencies (MAFs) >55% or *TP53/PIK3C/PTEN* MAFs equal to *KRAS/GNAS* MAFs, consequently sensitivity and specificity rose to 100%. Low-frequency mutations in *TP53/PIK3CA/PTEN* could also predict future malignant progression in low-risk cysts [16]. Interestingly, molecular analysis significantly outperformed cytopathology in this study, which argues that molecular evolution likely precedes histomorphologic changes and could potentially have the benefit of early detection of advanced neoplasia. Another group recently identified a *KLF4* mutation that was found to be significantly more prominent in low-grade lesions, thus offering an additional target for risk stratification in the future [33]. NGS holds significant potential for future use in risk stratification. It represents a significant financial barrier in the adoption of its use, but one must consider the total cost of unnecessary surgery and related complications when deciding on the course of clinical management.

### 3.2. DNA Quantity and Loss of Heterozygosity in Pancreatic Cyst Fluid

Cyst fluid molecular analysis has been found to be a valuable tool in risk stratification of PCLs and has even been shown to have efficacy nearly identical to micro forceps biopsy [34]. This could be due in part to cyst wall degeneration leading to the deposition of DNA into the cyst fluid. Therefore, it is important to mention the usefulness of analyzing DNA quantity and loss of heterozygosity in cystic fluid samples. Simpson et al. prospectively reviewed the medical records of over one thousand patients diagnosed with branch duct IPMN and concluded that the presence of an increased quantity of DNA in cyst fluid can predict (*p* = 0.004) a high risk of malignant transformation with a sensitivity and specificity of 78.3% and 52.7%, respectively [35]. These findings would likely support the hypothesis that malignant degeneration is followed by the sloughing of the cyst wall into the fluid-filled cavity [36]. Further investigation revealed high clonality loss of heterozygosity in tumor suppressor genes to be highly specific for advanced neoplasia, and when combined with DNA quantity analysis, the combination reached a sensitivity of 84.7% and specificity of 99%, respectively [35]. Analysis of these parameters along with other standard tests (namely fluid chemistry, imaging, and cytology) has led to the development of a malignancy risk scoring system referred to as the Integrated Molecular Pathology (IMP). The IMP was able to accurately identify malignant potential with significantly improved specificity compared to the 2012 International Consensus Guidelines (ICG) [37]. In a more recent retrospective study by Simpson et al., they coined the IMP-10 which combined the previously mentioned IMP with an additional component of a main pancreatic duct diameter of greater than 10mm. The IMP-10 was significantly more specific and accurate in the detection of invasive disease than either the original IMP or ICG [38].

### 3.3. Telomere Fusions

Telomeres are repetitive, non-coding DNA sequences located at the end of all human chromosomes that function to protect against chromosomal degradation and instability. Early studies have indicated telomere shortening to be representative of dysplasia in branch duct IPMNs [39]. The rapid cell division of neoplasia leads to telomere shortening and possible fusion. Telomere fusion-induced DNA damage can drive the progression of precancerous lesions [17]. In 2018, Hata et al., performed a telomere fusion assay on 93 pancreatic cyst fluid samples and found there to be telomere fusions in 0% of the low-grade branch duct IPMNs. Furthermore, they reported that the prevalence of telomere fusions increased with advancing histologic grade. The difference in the prevalence of fusions between advanced lesions and low-grade lesions was statistically significant (*p* = 0.025) [17]. Telomere fusion detection could be a helpful tool to provide supplementary information for risk-stratification in the future but currently is limited to research studies. 

### 3.4. MicroRNA

MicroRNAs (miRNA) are short, noncoding segments of RNA that function in the regulation of post-transcriptional gene expression. Past studies have examined whether the expression of specific miRNA could risk-stratify PCLs. In 2012, Matthaei et al., identified 37 miRNA that could risk-stratify branch duct IPMNs, nine of which also accurately predicted the need for surgical resection [40]. Building upon this, Lee et al., developed a panel of miRNAs (*miR-21-5p*, *miR-485-3p*, *miR-708-5p*, and *miR-375*) that distinguished between IPMN and pancreatic adenocarcinoma with a sensitivity of 95% and a specificity of 85% [41].

A recently published study by Shirakami et al., discovered 6 miRNAs (*miR-711*, *miR-3679-5p*, *miR-6126*, *miR-6780b-5p*, *miR-6798-5p*, and *miR-6879-5p*) to be significantly elevated (*p* < 0.05) in the pancreatic cyst fluid of cancerous branch duct IPMNs compared to noncancerous branch duct IPMNs [18]. This single-center study was limited by a small sample size, indicating that a larger scale study may be necessary to validate these results. There is clearly a role that miRNAs play in the progressions of branch duct IPMNs, but the true question is which miRNA signatures are the most accurate and reliable for risk-stratification. 

### 3.5. DNA Methylation

DNA methylation has been found to be present in almost every form of human cancer and it is responsible for the improper silencing of genes. Consequently, anomalous DNA hypermethylation at CpG islands, typically in promoter regulatory regions, leads to loss of gene function and tumor progression. In 2008, Hong et al., explored methylation as an avenue for risk-stratification by running PCR on seven genes commonly found to be methylated in pancreatic neoplasms (*SPARC*, *SARP2*, *TSLC1*, *RELN*, *TFPI2*, *CLDN5*, *UCHL1*). They found increasing levels of improper DNA methylation with increasing grade of the lesion [42]. Additional aberrant CpG island methylation patterns have been found more frequently in branch duct IPMNs with high-grade dysplasia than low-grade dysplasia, including in *BNIP3*, *PTCHD2*, *SOX17*, *NXPH1*, and *EBF3* by both genome-wide screens and individual methylation sequencing. Interestingly *BNIP3* methylation was not identified in any branch duct IPMNs with low-grade dysplasia [43].

More recently, Hata et al., analyzed 113 branch duct IPMN cyst fluid specimens for methylation patterns in seven previously identified commonly hypermethylated genes (*SOX17*, *PTCHD2*, *BNIP3*, *FOXE1*, *SLIT2*, *EYA4*, and *SFRP1*). Contrary to past findings mentioned above, this group found *BNIP3* to be the only marker out of the seven which did not differentiate advanced neoplasia. Hypermethylation in a combination of genes was higher in branch duct IPMN with high-grade dysplasia compared to low-grade lesions. They then examined the performance of specific markers alone and found methylation of the *SOX17* gene to have the highest diagnostic ability in detecting advanced neoplasia in branch duct IPMNs with a sensitivity and specificity of 83% and 81.8%, respectively [19]. When a combination of DNA methylation targets was used, diagnostic accuracy was 88% with an increase in sensitivity and specificity when compared to single marker analysis [19]. Majumder et al., discovered and validated a panel of methylated DNA targets in pancreatic cyst fluid that significantly outperformed CEA levels and *KRAS* mutation detection in the accurate identification of high-risk lesions. This study identified two methylated genes (*TBX15* and *BMP3*) that when analyzed together, could discriminate between high-grade and low-grade lesions with a sensitivity and specificity of 90% and 92%, respectively [20]. Notably, this study included analysis of multiple cyst types with branch duct IPMN being a significant proportion. These data support the use of DNA methylation as a molecular tool for risk-stratifying branch duct IPMN. Future studies need to focus on the most accurate of the many methylation targets for this to become applicable in clinical practice. 

**Table 2 diagnostics-12-02573-t002:** Overview of reviewed studies regarding the use of cyst fluid for risk stratification of branch duct IPMNs.

Author	Molecular Marker(s)	Key Findings	Diagnostic Parameters	Conclusions
Singhi (2018) [16]	TP53, PIK3CA, PTEN, KRAS, GNAS	Combination of KRAS/GNAS and TP53/PTEN/CDKN2A indicates advanced neoplasia	Analysis of KRAS/GNAS and TP53/PTEN/CDKN2A was 100% specific and 89% sensitive for advanced neoplasia	The integration of molecular testing in pre-operative patients can be useful in predicting future risk of malignancy
Hata (2018) [17]	Telomere fusions	Branch duct IPMN cyst fluid aspirates revealed no telomere fusions in low grade lesions, however prevalence increased with advancing histologic grade	Prevalence of telomere fusions: Low grade 0% Intermediate 15.4% High 26.9% Cancer 42.9% *p* = 0.025	Telomere fusions can be readily detected in cyst fluid and are helpful in predicting the grade of dysplasia
Shirakami (2021) [18]	miR-711, miR-3679-5p, miR-6126, miR-6780b-5p, miR-6798-5p, and miR-6879-5p	Six miRNAs were significantly elevated in the cyst fluid of branch duct IPMN with carcinoma when compared to benign branch duct IPMNs	Differences in miRNA levels between low-grade and high-grade lesions were all statistically significant (*p* < 0.05)	Certain miRNAs are elevated in the cyst fluid of cancerous lesions thus offering the potential to predict high risk lesions requiring surgical resection
Hata (2017) [19]	SOX17, PTCHD2, BNIP3, FOXE1, SLIT2, EYA4, and SFRP1	Gene methylation patterns can accurately distinguish between advanced neoplasia and low-grade lesions (all but BNIP3)	Single marker: SOX17 sensitivity 83%, specificity 82% Two gene: SOX17/FOXE1 or EYA4 accuracy 86% Four gene: FOXE1/SLIT2/EYA4/SFRP1 accuracy 88%, 84% sensitivity, and 89% specificity	Cyst fluid analysis of gene methylation patterns, whether single gene or in combination, can accurately distinguish between advanced neoplasia and low-grade lesions
Majumder (2019) [20]	TBX15 and BMP3	Two gene methylation analysis can discriminate between advanced neoplasia and low-grade lesions	Combination of methylated TBX15 and BMP3 had sensitivity 90% and specificity of 92% for detecting advanced neoplasia	Methylation analysis of this two gene combination can be useful in predicting grade of dysplasia
Singhi (2016) [26]	TP53, PIK3CA, PTEN	Molecular analysis of cyst fluid was able to detect advanced neoplasia via mutations in TP53, PIK3CA, and/or PTEN	Detected branch duct IPMN harboring advanced neoplasia with 91% sensitivity and 97% specificity	Integration of molecular analysis in PCLs can better detect cysts with advanced neoplasia than AGA guidelines
Khalid (2009) [29]	KRAS, allelic loss, DNA quantity	10/40 malignant cysts had negative cytology, all of which could be diagnosed as malignant with high amplitude KRAS mutation in conjunction with high amplitude allelic loss	High amplitude KRAS mutation followed by allelic loss: 96% specific and 37% sensitive for malignancy in the cyst	Increased cyst fluid DNA quantity, high-amplitude mutations, and allelic loss can be used to predict malignancy, especially when cytology is negative
Springer (2015) [30]	SMAD4, LOH in RNF43 and TP53, Chromosomal aneuploidy	Analysis for SMAD4, TP53, LOH in chromosome 17, or aneuploidy of 5p, 8p, 13q, or 18q could correctly identify high-grade dysplasia or invasive cancer. This could reduce unnecessary operations by 91%	The panel could identify patients requiring surgery with 75% sensitivity and 92% specificity	Molecular analysis of cyst fluid can be used to risk-stratify cysts with malignant potential and can reduce the amount of unnecessary operations
Rosenbaum (2017) [32]	TP53, SMAD4, CDKN2A, NOTCH1	NGS revealed mutations in TP53, SMAD4, CDKN2A, and NOTCH1 to only be present within malignant cysts.	These mutations had 100% specificity and 46% sensitivity for carcinoma	Variants in TP53, SMAD4, CDKN2A, and NOTCH1 favor the diagnosis of high-risk cyst and warrant surgery or further investigation
Fujikura (2021) [33]	KLF4	KLF4 mutations detected in cyst fluid samples were significantly more prevalent in cysts with low grade dysplasia	KLF4 prevalence: Low grade 50% Intermediate 39% High 15%	High and low grade IPMNs have distinct molecular pathways with KLF4 mutations being enriched in the low grade pathway
Simpson (2018) [35]	DNA quantity, LOH in tumor suppressor	High DNA quantity in conjunction with high clonality LOH in tumor suppressor genes could detect advanced neoplasia	High quantity DNA and LOH had specificity of 99%, PPV 60%, and diagnostic accuracy of 91% for advanced lesions	Increased DNA quantity along with LOH in tumor suppressors can be predictive of high-risk lesions

### 3.6. Long Non-Coding RNA as a Potential Future Target for Cyst Fluid Molecular Analysis

Long non-coding RNAs (lncRNA) play a significant role in regulating gene expression, though they do not produce any protein themselves [44]. Specific lncRNAs have been found to contribute to the progression, growth, and metastasis of solid pancreatic cancers, as well as other malignancies [45,46]. Various studies have examined whether levels of specific lncRNAs can be correlated with the progression of pancreatic malignancy. Some have investigated the function of lncRNAs as potential tumor suppressors in pancreatic cancer. In 2019, Yue and Guo found that the specific lncRNA TUSC-7 was present at lower levels in lesions that progressed to pancreatic cancer, and also found that lesions with low levels of TUSC-7 had a lower survival rate compared to those with higher levels [47]. In addition, Zhang et al., described the ability of another lncRNA, CASC2, to inhibit the migration of malignant cells by inhibiting MiR-21 [48]. Both studies showed that lncRNAs could have suppressive effects on tumor growth and spread, which is suggestive that they do indeed have some level of tumor suppressor function. However, none of them examined whether increased levels of different lncRNAs were predictive of the presence or prognosis of pancreatic cancer in premalignant lesions. More specifically to IPMNs, Permuth et al. found a panel of eight lncRNAs that were able to distinguish between IPMNs that needed surgical resection and those that did not at a higher rate than the standard clinical and radiologic features which had been used previously [49]. Most recently, another study discovered one lncRNA that was predictive of IPMN tumorigenesis on its own (lncRNA-TFG) and two lncRNAs that were associated with poor survival (HAND2-AS1 and CTD-2033D15.2) [50]. These studies were not completed specifically on branch duct IPMN fluid samples; however, the abundance of data on lncRNA and its implication in the carcinogenesis pathway warrants future investigation within branch duct IPMN cyst fluid samples. If lncRNAs do play a prominent role in the progression to pancreatic cancer as theorized above, identification of these markers in cyst fluid could be a promising additional biomarker that can be utilized for risk stratification.

## 4. Conclusions

Molecular analysis of pancreatic cyst fluid has proven to be a valuable tool that can be used in conjunction with current clinical management to improve the classification and risk stratification of branch-duct IPMNs, however, molecular analysis has yet to be proven to be reliable as the sole diagnostic method for these lesions. This is in part because research on this topic requires a pathologic diagnosis of cyst type and grade of dysplasia, thus limiting the ability of researchers to obtain a sufficient patient population from which to draw definitive conclusions. Secondly, surgical resection is often pursued based on concerning clinical and radiographic features. Asymptomatic patients with no radiologic indication for surgery are unable to have their cyst type and grade of dysplasia confirmed without putting them at risk of unnecessary surgery. Furthermore, the outcomes of non-surgical cysts are often unknown, and are difficult to study. In the absence of other surgical indications, these patients would benefit the most from a highly sensitive and specific molecular panel to accurately predict malignancy risk. Review of the studies mentioned above suggest that there are a variety of different measurable factors in the cyst fluid that can be used to aid in the diagnosis of cyst type and grade of dysplasia along with standard imaging, serologies, and clinical features. Larger studies are needed to highlight which of the many options is the most cost-effective, accurate, and easily performed in order to improve the diagnosis and treatment of branch duct IPMNs. One current problem limiting clinical implementation is the lack of widespread availability of technology needed to perform molecular analysis panels. Although the cost of DNA analysis can be variably high, this may be offset by a reduction in surgical costs. Implementation of a reliable method to risk stratify IPMNs will result in a lower number of unnecessary pancreatic resection surgeries and the morbidity that comes with them. While reimbursement for molecular studies remains a challenge, the advancement of technology and ever-decreasing cost as equipment becomes cheaper make it promising for more widely available testing in the future. 

## Figures and Tables

**Table 1 diagnostics-12-02573-t001:** Overview of studies investigating the diagnostic utility of KRAS and GNAS in branch duct IPMN.

Author	Molecular Marker(s)	Key Findings	Diagnostic Parameters	Conclusions
Ren (2021) [15]	BRAF, KRAS, GNAS	In 60% of KRAS-/GNAS+ branch duct IPMNs, alternate BRAF mutations were present	BRAF variant allele frequencies: 15.7–46.9% GNAS variant allele frequencies: 21.3–50.5%	Somatic BRAF mutations are indicated in the carcinogenesis of KRAS negative branch duct IPMNs
Singhi (2018) [16]	KRAS, GNAS, CEA	Next generation sequencing detected KRAS/GNAS mutations in 100% of branch duct IPMNs	KRAS and/or GNAS detection was 100% sensitive and 96% specific for branch duct IPMN	KRAS and/or GNAS mutations are highly sensitive and specific for branch duct IPMN identification
Singhi (2014) [25]	KRAS, GNAS	GNAS/KRAS mutations aid in mucinous differentiation and identification of branch duct IPMNs	Branch duct IPMN identification: Sensitivity 84% and Specificity 98% Mucinous differentiation: Sensitivity 65% and Specificity 100%	Multi-gene analysis of GNAS and KRAS was highly sensitive and specific for branch duct IPMN identification
Singhi (2016) [26]	KRAS, GNAS	KRAS and/or GNAS were identified in all 23 IPMNs	The sensitivity and specificity of KRAS and/or GNAS for IPMN reached 100%	KRAS and GNAS mutational analysis is very sensitive and specific for branch duct IPMN diagnosis
Kadayifci (2017) [27]	KRAS, GNAS, CEA	GNAS/KRAS mutations are specific for branch duct IPMNs but have low diagnostic accuracy however diagnostic accuracy improved when both are used in conjunction with CEA	Diagnostic accuracy (%): KRAS 76.6 GNAS 70 KRAS/GNAS 80.7 KRAS/GNAS/CEA 86	KRAS and GNAS mutations along with elevated CEA is accurate for branch duct IPMN diagnosis

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
