# Peer review of "Molecular Analysis of Pancreatic Cyst Fluid for the Management of Intraductal Papillary Mucinous Neoplasms"

_diagnostics, 2022, doi:10.3390/diagnostics12112573_

Round 1

Reviewer 1 Report

The Authors reviewed the role of cyst fluid molecular analysis and their utility in the diagnosis and risk stratification of branch duct IPMNs. The reiew is interesting,  well-written and shows an old problem: the correct diagnosis of malignant IPMNs, avoiding operation for benign disease. Does molecular analysis of cyst fluid solve this problem? I think it is difficult to answer yes. In all experiences, the number of patients who underwent resection is too small to draw definitive conclusions and the indication for operation was based presumably on the well-established clinical or radiological criteria. In asymptomatic patients, whithout worrisome or high-risk features, does molecular analysis justify pancreatic resection? Moreover, not all laboratories are able to perform this analysis, the cost of such procedures is  not clear, the follow-up is too short, and the outcome of non operated cysts is unknown. I think these points require more extensive discussion.

Reviewer 2 Report

The review is pertinent and relevant.

The discussion needs to be enhanced. The following references are suggested to discussed and subsequently added 

1. Arner DM, Corning BE, Ahmed AM, et al. Molecular analysis of pancreatic cyst fluid changes clinical management. Endosc Ultrasound. 2018;7(1):29-33. doi:10.4103/eus.eus_22_17

2. Ohtsuka T, Tomosugi T, Kimura R, Nakamura S, Miyasaka Y, Nakata K, Mori Y, Morita M, Torata N, Shindo K, Ohuchida K, Nakamura M. Clinical assessment of the GNAS mutation status in patients with intraductal papillary mucinous neoplasm of the pancreas. Surg Today. 2019 Nov;49(11):887-893. doi: 10.1007/s00595-019-01797-7. Epub 2019 Mar 16. PMID: 30879148.

3. Schmitz D, Kazdal D, Allgäuer M, Trunk M, Vornhusen S, Nahm AM, Doll M, Weingärtner S, Endris V, Penzel R, Kirchner M, Brandt R, Neumann O, Sültmann H, Budczies J, Kienle P, Magdeburg R, Hetjens S, Schirmacher P, Bergmann F, Rudi J, Stenzinger A, Volckmar AL. KRAS/GNAS-testing by highly sensitive deep targeted next generation sequencing improves the endoscopic ultrasound-guided workup of suspected mucinous neoplasms of the pancreas. Genes Chromosomes Cancer. 2021 Jul;60(7):489-497. doi: 10.1002/gcc.22946. Epub 2021 Mar 16. PMID: 33686791.

4. Jones M, Zheng Z, Wang J, Dudley J, Albanese E, Kadayifci A, Dias-Santagata D, Le L, Brugge WR, Fernandez-del Castillo C, Mino-Kenudson M, Iafrate AJ, Pitman MB. Impact of next-generation sequencing on the clinical diagnosis of pancreatic cysts. Gastrointest Endosc. 2016 Jan;83(1):140-8. doi: 10.1016/j.gie.2015.06.047. Epub 2015 Aug 5. PMID: 26253016.

5. Laquière AE, Lagarde A, Napoléon B, Bourdariat R, Atkinson A, Donatelli G, Pol B, Lecomte L, Curel L, Urena-Campos R, Helbert T, Valantin V, Mithieux F, Buono JP, Grandval P, Olschwang S. Genomic profile concordance between pancreatic cyst fluid and neoplastic tissue. World J Gastroenterol. 2019 Sep 28;25(36):5530-5542. doi: 10.3748/wjg.v25.i36.5530. PMID: 31576098; PMCID: PMC6767987.

6. McCarty TR, Paleti S, Rustagi T. Molecular analysis of EUS-acquired pancreatic cyst fluid for KRAS and GNAS mutations for diagnosis of intraductal papillary mucinous neoplasia and mucinous cystic lesions: a systematic review and meta-analysis. Gastrointest Endosc. 2021 May;93(5):1019-1033.e5. doi: 10.1016/j.gie.2020.12.014. Epub 2021 Mar 26. PMID: 33359054.

Round 2

Reviewer 1 Report

The Authors adequately replied to my comments. The paper is suitable for publication